# Collagen V α1 Chain Decrease in Papillary Dermis from Early Systemic Sclerosis: A New Proposal in Cutaneous Fibrosis Molecular Structure

**DOI:** 10.3390/ijms232012654

**Published:** 2022-10-21

**Authors:** Jymenez de Morais, Ana Paula P. Velosa, Priscila C. Andrade, Denise Frediani, Solange Carrasco, Zelita A. de Jesus Queiroz, Patrícia Martin, Renata F. Saito, Vitória Elias, Cláudia Goldenstein-Schainberg, Roger Chammas, Percival D. Sampaio-Barros, Vera L. Capelozzi, Walcy R. Teodoro

**Affiliations:** 1Division of Rheumatology (LIM 17), Faculdade de Medicina da Universidade de São Paulo, São Paulo 01246-903, Brazil; 2Department of Pathology, Faculdade de Medicina da Universidade de São Paulo, São Paulo 01246-903, Brazil; 3Discipline of Clinical Emergency (LIM 51), Faculdade de Medicina da Universidade de São Paulo, São Paulo 01246-903, Brazil; 4Center for Translational Research in Oncology (LIM24), Department of Radiology and Oncology, Faculdade de Medicina da Universidade de São Paulo, São Paulo 01246-903, Brazil; 5Instituto do Câncer do Estado de São Paulo, São Paulo 01246-903, Brazil

**Keywords:** collagen V alpha chains, skin, systemic sclerosis, papillary dermis, fibrosis

## Abstract

Cutaneous fibrosis is one of the main features of systemic sclerosis (SSc). Recent findings correlated abnormal collagen V (Col V) deposition in dermis with skin thickening and disease activity in SSc. Considering that Col V is an important regulator of collagen fibrillogenesis, understanding the role of Col V in the first two years of the skin fibrosis in SSc (early SSc) can help to determine new targets for future treatments. In this study, we analyzed the morphological, ultrastructural and molecular features of α1(V) and α2(V) chains and the expression of their coding genes *COL5A1* and *COL5A2* in collagen fibrillogenesis in early-SSc. Skin biopsies were obtained from seven consecutive treatment-naïve patients with SSc-related fibrosis and four healthy controls. Our data showed increased α1(V) and α2(V) chain expression in the reticular dermis of early-SSc patients; however, immunofluorescence and ultrastructural immunogold staining determined a significant decreased expression of the α1(V) chain along the dermoepidermal junction in the papillary dermis from early-SSc-patients in relation to the control (12.77 ± 1.34 vs. 66.84 ± 3.36; *p* < 0.0001). The immunoblot confirmed the decreased expression of the α1(V) chain by the cutaneous fibroblasts of early-SSc, despite the increased *COL5A1* and *COL5A2* gene expression. In contrast, the α2(V) chain was overexpressed in the small vessels (63.18 ± 3.56 vs. 12.16 ± 0.81; *p* < 0.0001) and capillaries (60.88 ± 5.82 vs. 15.11 ± 3.80; *p* < 0.0001) in the reticular dermis of early-SSc patients. Furthermore, COLVA2 siRNA in SSc cutaneous fibroblasts resulted in a decreased α1(V) chain expression. These results highlight an intense decrease in the α1(V) chain along the dermoepidermal junction, suggesting an altered molecular histoarchitecture in the SSc papillary dermis, with a possible decrease in the expression of the α1(V)3 homotrimeric isoform, which could interfere with the thickening and cutaneous fibrosis related to SSc.

## 1. Introduction

Systemic sclerosis (SSc) is a chronic autoimmune disease in which the fibrosis process is particularly known to contribute to the significant loss of tissue function [1,2,3]. As SSc patients with a worse prognosis have a predominantly severe skin involvement [4], an early detection of those patients who will develop severe disease is an important therapeutic strategy [1]. The process of cutaneous fibrosis in SSc is characterized by the accumulation of type I, III and V collagens assembled in a single macromolecule to form heterotypical fibers. Particularly, collagen V (Col V), which represents only 3 to 5% of the total collagen in the cutaneous tissue, can occur in different isoforms. The most abundant α1(V)2, α2(V) heterotrimer control type I/V collagen heterofibrils and small amounts of the α1(V)3 homotrimer serve as a bridging molecule at the epidermis–dermis interface [5,6].

The gene expression in SSc skin has shown striking differences from the gene expression in healthy controls along with specific gene expression signatures [3,7,8]. In a previous study by our group, an increased expression of thickened Col V and the messenger RNA of *COL5A2* gene in patients with early SSc-related cutaneous fibrosis was found [9]. The same genetic profile was observed in patients with end-stage SSc-related interstitial lung disease, associated with reduced vital capacity and diffusion capacity for carbon monoxide, indicating that this protein can play an important role in the lung tissue in patients with SSc [10]. These results suggest that Col V may be involved in the SSc pathogenesis and gain support in an animal model showing that the immunization of C57/BL6 mice with this protein reproduces the major clinical and morphological features of SSc [11]. 

Recently, numerous studies that analyzed gene expression revealed important pathways in the pathogenesis of SSc, focusing on the differentiation of clinical subtypes [12] and disease stages [13], as well as on the evaluation of their reproducibility in animal models [14]. So, the present study was undertaken to gain insight into potential pathogenic pathways which can be important for patients in the first two years of disease, which is considered as a key moment in the treatment of SSc cutaneous fibrosis and considered as early-SSc in this study. This study will analyze the hypothesis that the morphological, ultrastructural and molecular features of α1(V) and α2(V) chains may represent key elements to the pathogenesis of this severe complication, as an increased expression of Col V was observed primarily in patients who developed cutaneous fibrosis.

## 2. Results

### 2.1. The Col V Alpha 1 Chain in Papillary Dermis Exhibits a Downregulation Pattern in Early SSc-Related Cutaneous Fibrosis 

The analysis of the sections stained with H&E and Masson trichrome revealed a hypertrophic SSc epidermis with a hypergranular layer, hyperkeratosis and elongation of the epithelial cone covering and nonhomogeneous fibrillary matrix, with the disarrangement of thin and thick collagen fibers when compared to the healthy control skin (Figure 1A). As previously shown [9], there was the expression of Col V in the skin of patients with early SSc, but changes in the different isoforms of this protein, which are important in Col V fibrillogenesis and skin structure, were not evaluated. In this way, the sections immunostained in Figure 1B show the α1(V) and α2(V) chain localization in the papillary and reticular dermis, and the sections in Figure 1C show the alpha V chain colocalization using three-dimensional reconstruction in the SSc patient and health control skin. The nuclei were stained blue with DAPI. In the healthy dermis, an intense staining of the α1(V) chain was observed along the dermoepidermal junction spreading to the small vessels of the papillary dermis (Figure 1B,C). In contrast, the α1(V) expression was significantly decreased along the dermoepidermal junction in the papillary dermis layer of the SSc patients (Figure 1B–D). On the other hand, α1(V) and α2(V) expression increased in the reticular dermis layer of the SSc skin compared to the control (Figure 1B–D). By this time, the α2(V) expression was increased in the SSc capillaries (Figure 1B–D) and vessels of the reticular dermis (Figure 1B–D). The inhomogeneous distribution and fluorescence of the α1(V) and α2 (V) chains in the fibrillary matrix of the SSc skin coincided with the different area fraction occupied in the papillary dermis, small vessels and reticular dermis. The area fraction occupied by the α1(V) chain in the papillary dermis was significantly decreased in the SSc compared to the healthy control (12.77 ± 1.34 vs. 66.84 ± 3.36; *p* < 0.0001), whereas the area fraction of the α2(V) chain was similar in both groups (Figure 1D). In contrast, the area fraction of the α1(V) and α2(V) chains in the fibrillary matrix was significantly higher in the reticular dermis of the SSc than the healthy control (Figure 1D, α1(V): 13.75 ± 0.96 vs. 7.66 ± 0.2133; *p* < 0.001; α2(V): 21.07 ± 0.790 vs. 5.072 ± 0.4117; *p* < 0.0001). The area fraction occupied by the α2(V) chains along the capillaries (60.88 ± 5.817 vs. 15.11 ± 3.80; *p* < 0.0001) and wall of the vessels (63.18 ± 3.56 vs. 12.16 ± 0.811; *p* < 0.0001) was significantly higher in the SSc skin than the healthy skin (Figure 1D), while the area fraction occupied by the α1(V) chain was similar in both groups.

Using electron microscopy, the healthy skin presented homogeneous collagen fibrils resting in the basal membrane (BM) and keratinocytes (Ke), beyond the normal delimitation of the BM, papillary dermis (PD), reticular dermis (RD) and normal collagen fibril architecture (Figure 2A–D). In contrast, the SSc skin exhibited rectification of the epidermis (Epi) and fusion of the BM with PD (Figure 2E,F) and the disarray and prominence of collagen (Col) thick fibrils determining significant thickness (Figure 2G,H). The healthy skin presented an homogeneous fibrillary matrix uniform arrangement of thin and thick collagen fibers (Figure 2C,D), coincident with the strong immunogold staining of α1(V) in the microfibrils along the papillary dermis (Figure 2Ca, insert in 2C) and the weak immunogold staining of α1(V) and α2(V) in the reticular dermis (Figure 2Da,Db, insert in Figure 2D). In contrast, the SSc epidermal–dermal junction showed a decreased immunogold staining of α1(V) (Figure 2Ga, insert in 2G) compared to the control (Figure 2Ca, insert in Figure 2C), corroborating the findings in the immunofluorescence (Figure 1B–D). A strong immunogold staining of α1(V) and α2(V) is shown joining the thickened region of the depth dermis (DD) (Figure 2Ha,Hb, insert in Figure 2H) from the SSc skin in contrast with the control (Figure 2Da,Db, insert in Figure 2D).

### 2.2. SSc Cutaneous Fibroblasts Express Decreased Col V Alpha 1 Chain despite the Overexpression of COL5A1 Gene

The relative gene expression of the cutaneous fibroblasts showed a significant increase in *COL5A1* and *COL5A2* for the SSc skin compared to the healthy controls (2.07 ± 0.31 vs. 0.54 ± 0.10, *p* = 0.01 and 1.40 ± 0.15 vs. 0.19 ± 0.03; *p* = 0.002, respectively) (Figure 3A). In addition, the immunoblot analysis showed a significant lower relative expression of the α1(V) chain produced by the cutaneous fibroblasts from SSc skin than the healthy controls (*p* < 0.05) (Figure 3B,C). Although the α2(V) chain produced by the cutaneous cells from SSc patients was higher than the α1(V) chain, this difference was not significant. 

### 2.3. COLVA2 siRNA in SSc Cutaneous Fibroblasts Results in Decreased Col V Alpha 1 Chains Expression

Considering our findings showing increased α2(V) chain expression in the reticular dermis layer, capillaries and vessels of the SSc patient skin, we employed temporary inhibition of *COL5A2* RNA in the SSc cutaneous fibroblasts. As expected, we found a significant decrease in the mRNA *COL5A2* expression in the SSc cells cultured with *COL5A2* siRNA transfection compared to the mRNA COLVA2 in the SSc fibroblasts not transfected (0.19 ± 0.0182 vs. 1.40 ± 0.15; *p* < 0.0001) (Figure 4A). Although there was an increase in the expression of mRNA *COL5A1* by the SSc cells after COLVA2 siRNA, this difference did not reach statistical significance (Figure 4A). Figure 4B shows the evaluation of the three-dimensional reconstruction of the partial distribution of the α1(V) and α2(V) chains without interference and after the temporary inhibition of *COL5A2*. The α1(V) chain from the SSc cultured cells exhibited prominent arrays of filamentous intracytoplasmic. After the *COL5A2* siRNA of the SSc cultured cells, the α1(V) chain expression decreased and showed a pattern of fine fibers distributed throughout the cell. In contrast, the α2(V) chain from the SSc cells was overexpressed, especially around the nucleus and throughout the cytoplasm. As expected, after the temporary inhibition of *COL5A2*, the expression of the α2(V) chain by the SSc cells decreased.

## 3. Discussion

Considering the ample evidence that molecular isoforms of the same type of collagen can differentially affect the architecture of the dermal matrix and that the altered collagen deposition leads to a loss of function, in the present study, we suggest that morphological, ultrastructural and molecular features of the α1(V)3 homotrimer isoform can be associated with skin thickening in the cutaneous fibrosis of early-SSc. For this reason, we chose to select only patients who were diagnosed with SSc within the first two years since the first non-Raynaud’s symptom and who were treatment-naïve without immunosuppressors and/or antifibrotics. 

Our data set suggest that the atypical α1(V) chain low expression along the dermoepidermal junction, associated with high expression of α1(V) and α2(V) chains in reticular dermis, results in biomechanical consequences, contributing to the rigidity of the skin in early-SSc. It is worth mentioning that Col V has homotrimer and heterotrimer isoforms, composed of α1(V) and α2(V) chains [15,16,17]. In skin, the Col V homotrimer isoform, formed by three α1(V) chains [α1(V)3], forms microfibrils that bind to large collagen I fibrils and play the role of bridging the epidermis–dermis interface [6,16]. By this time, the heterotrimer isoform formed by the two α1(V) chains and one α2(V) chain [α1(V)2 α2(V)] was found inside the mixed collagen I/III fibrils and had the function of regulating the diameter of these heterotypic fibrils [15,18]. In fact, the α1(V) chain decrease in the dermoepidermal junction shown in the early-SSc patient skin significantly contrasted with the higher content of Col V characteristic of the papillary dermis from healthy skin [19]. Furthermore, as expected and corroborating with our previous work, the α1(V) and α2(V) increase in the reticular dermis suggests Col V heterotrimer isoform [α1(V)2 α2(V)] high expression is associated with the onset of skin thickening in early-SSc [9]. 

Of note, the present study is the first to describe the altered fibrillar matrix in the papillary dermis related to α1(V) chain low expression along the dermoepidermal junction shown by immunofluorescence and ultrastructural immunogold staining. Our findings suggest that the irregular distribution of the α1(V)3 homotrimer Col V isoform may have a pivotal role in skin biomechanical alteration, contributing to the rigidity and skin thickening in early SSc-related cutaneous fibrosis. This hypothesis is supported by the fact that the α1(V) chain forms thin filaments of the molecular α1(V)3 isoform, which acts as a bridge molecule at the dermoepidermal interface, along with the collagen IV and laminin present in the basement membrane on healthy skin, thereby contributing to the stabilization of the epidermal–dermal interface. In fact, a transgenic mouse model of Ehlers–Danlos syndrome demonstrated that the overexpression of the α1(V)3 isoform in the dermal–epidermis interface modifies the skin biomechanical properties by altering the skin deformability potential to hyperextensible [6]. Based on these data, we hypothesized that the loss of α1(V)3 chain expression across the dermoepidermal interface could explain the skin thickening in early-SSc, contrasting with the skin hyperdistensibility in Ehlers–Danlos syndrome [20]. Inferring our findings, we can assume that the significant decrease in the α1(V) in the dermoepidermal region could play a role in the skin tightening that occurs in the first two years of SSc. 

Interestingly, we showed that the increased levels of mRNA *COL5A1* did not result in an increased α1(V) chain expression by cutaneous fibroblasts from early-SSc patients. This finding is further evidence that early-SSc patient skin undergoes an alteration in the fibrillogenesis of the Col V isoforms/subtypes. It is already known that although the fibroblasts produce the extracellular matrix present in cutaneous tissue, these cells have distinct proliferative capacities, morphological characteristics and cell surface markers in the papillary and reticular dermis regions [21,22]. Furthermore, other studies have indicated that undifferentiated cells are localized in the papillary dermis and synthesize Col V to develop a unique niche and that Col V plays an important role in maintaining the undifferentiated state of these cells [19]. Furthermore, when normal human fibroblasts were cultured with Col V but not with other collagen types, they expressed markers of an undifferentiated state [19]. Based on these findings, we suggest that the decreased α1(V) expression by SSc cutaneous fibroblasts along the dermoepidermal junction in the papillary dermis layer of early-SSc patients could result from a cellular differentiation in early SSc-related cutaneous fibrosis. In fact, although we have not investigated, we believe that the increased expression of TGF-β1, a well-known profibrotic cytokine found in the early stages of SSc, can contribute to induce the differentiation of papillary fibroblasts to a phenotype of reticular fibroblasts, which expresses increased alpha smooth muscle actin-α (α-SMA) and fibrillar type I and III collagen [23,24,25]. In fact, the skin samples from early-SSc patients involved in the present study shared the same histological diagnosis of mild fibrosis (Table 1), where an inflammatory profile with a greater amount of TGF in relation to the sclerotic stage is expected [13,26]. Differentiated fibroblasts and myofibroblasts synthesized large amount of fibrillar type I and III collagen that are key actors in dermal/epidermal-associated fibrosis, defined as a fibrotic state characterized by an excessive synthesis, deposition and remodeling of fibrillar collagens [9,27,28]. It, in turn, leads to a fibrotic process of the skin that is a major player in the progression of many fibrotic diseases, including SSc and idiopathic pulmonary fibrosis [29,30].

It is important to point out that in the reticular dermis of early SSc skin, we showed an increased α1(V) and α2(V) collagen concomitant with increased fibril density on ultrastructural examination, which revealed irregular and loosely compacted collagen fibrils and the presence of typical “cauliflower” fibrils, which represent the histological hallmark of the disturbed fibrillogenesis of heterotypic collagen fibrils, corroborating with our previous study that showed an increased Col V deposition in the dermis of SSc early-stage [9]. Of note, in the SSc capillaries and vessels, the increase in the area occupied by the α2(V) chain was disproportionate to that stoichiometrically expected for the heterotrimer isoform [α1(V)2 α2(V)] of Col V. These data could be explained by the complex collagen biosynthesis involving enzymes and chaperones, responsible for the crosslinking and fixation of carbohydrates in the helical sequences of the collagen triple helix [31]. Although new studies are necessary to try to understand the mechanisms involved in this finding, we could speculate that the overexpression of the α2(V) chain may be related to an alteration in the lysine hydroxylation in the helical domains of the triple helix of Col V [32].

Considering the α2(V) chain increasing in the SSc skin vessels and the possibility of collagen biosynthesis alteration, we can infer that fibrotic skin in early-SSc can be different from fibrosis repair, making the dermis rigid and promoting an increased expression of adhesion molecules, as the α2(V) chain is rich in RGD sequences which bind collagen to adhesion molecules, such as integrins [33]. Interestingly, the COLVA2 siRNA in the SSc cutaneous fibroblasts, beyond the expected decrease in the α2(V) chain, also resulted in a decreased α1(V) chain expression by these fibroblasts, reinforcing the idea that the fibrillar histoarchitecture of Col V depends on the folding of three alpha chains determined by chain stoichiometry [34]. The significant decrease in the α1(V) chain after cutaneous SSc fibroblast transfection with siRNA *COL5A2* provides a rationale for the future development of a gene therapy that would have the potential for the transcriptional inhibition of mRNA to Col V alpha chains which is a possible therapeutic strategy in the treatment of SSc. The siRNA has already been used as a potential gene therapy in experimental models of pulmonary fibrosis and cancer metastasis regression [35,36,37]. However, it is necessary to evaluate in experimental models of SSc whether siRNA could reverse the thickening skin process present in humans, which was a limitation of our study. 

Unfortunately, one of the limitations of our study is the small number of skin samples from the patients evaluated. However, it is important to mention that skin samples from treatment-naïve patients in the early stages of SSc are not so easily found in medical practice. Despite this, our findings suggest that incorporating the expression profile of Col V chains into the routine molecular examination of biomarkers may help to indicate the stage of the fibrosis in SSc skin and may be a promising tool for selecting and personalizing therapy. However, to determinate the fibroblast phenotype in the papillary and reticular dermis, such as to evaluate the relationship of the fibroblast profile with the expression of the α1(V) and α(V) chains and consequently, the Col V subtype in the skin tissue in the early-SSc stage, is a limitation of the present study. In addition, the evaluation of the role of the Col V chains in the mechanisms of fibroblast differentiation in different regions of the skin tissue in early-SSc is a subject that needs to be explored. 

In summary, our study shows an intense decrease in the α1(V) chain along the dermoepidermal junction, suggesting an altered molecular histoarchitecture in the SSc papillary dermis, with a possible decreased expression of the α1(V)3 homotrimeric isoform, which could interfere with the thickening and cutaneous fibrosis related to early-SSc. Based on our findings, we infer that early fibrotic SSc skin is a collagenopathy that leads to an atypical fibrillar synthesis of Col V that may represent the first alteration in SSc skin. In addition, the results presented here provide important molecular evidence that Col V isoform profiles are involved in early fibrotic scleroderma skin with atypical fibrillogenesis modulated by the dermoepidermal junction. Considering that atypical expression of the α1(V) chain may interfere with skin thickening, we can infer that α1(V) chain expression regulation could be a possible therapeutic target at the onset of cutaneous fibrosis in SSc. 

Preclinical studies are therapeutically warranted to target up-regulated fibrogenic collagen chains, particularly in early fibrotic SSc skin. Furthermore, our results support the hypothesis that Col V is an important molecule that, in addition to regulating the diameter of heterotypic fibers, can interfere with fibroblast differentiation, and its qualitative and/or quantitative modification may interfere with the fibrillogenesis process.

## 4. Materials and Methods

### 4.1. Study Participants

Skin biopsy samples were obtained from the dorsal surface of the distal forearm of 7 patients involved in previous studies [9], in the first two years of the skin fibrosis in SSc from first non-Raynaud symptom and were treatment-naïve (early SSc). Patients were classified as SSc according to the American College of Rheumatology/European League against Rheumatism criteria [9,38]. Skin thickening was assessed by one trained rheumatologist using the Modified Rodnan Skin Score (MRSS) [9,39], and SSc activity was calculated using the Valentini Disease Activity Index (Table 1) [9,40]. Skin tissues from 4 healthy control age-and sex matched volunteers were obtained from the same region of patients in elective surgery procedures. Fibroblast dermal cell cultures were performed from all patients and control subjects. In accordance with the Declaration of Helsinki, all patients and controls gave informed consent for their participation in the study. The study was approved in 2010 year by the local ethics committee of HCFMUSP (CAPPesq 0331/10).

### 4.2. Histology

The skin samples were fixed in 10% formalin, embedded in paraffin, and 3 to 4 μm sections were stained with hematoxylin and eosin (H&E) and Masson’s trichrome.

### 4.3. Immunofluorescence and Confocal Microscopy

SSc and control skin sections were mounted in slides with 3-aminopropyltriethoxysilane (Sigma Chemical Co., St. Louis, MO, USA), dewaxed in xylol and hydrated in graded ethanol. Antigen retrieval was accomplished using enzymatic treatment with pepsin from porcine gastric mucosa (10,000 dry unit/mL) (Sigma Chemical Co., St. Louis, MO, USA) in acetic acid buffer at 0.5 N for 30 min at 37 °C. Nonspecific sites were blocked with 5% bovine serum albumin (BSA) in phosphate buffer saline (PBS) for 30 min at room temperature. The skin specimens were incubated overnight at 4 °C with mouse polyclonal antihuman-α1(V) chain (1:300) and rabbit polyclonal antihuman-α2(V) chain (1:80) (Sigma Chemical Co., St. Louis, MO, USA). After, the skin sections were washed in PBS with Tween20 0.05% and incubated for 60 min at room temperature with Alexa 488-conjugated goat anti-mouse IgG (1:200, Invitrogen, Eugene, OR, USA) and Alexa 488-conjugated goat anti-rabbit IgG (1:200, Invitrogen, Eugene, OR, USA). For negative and autofluorescence controls, sections were incubated with PBS and normal rabbit or mouse serum instead of the specific antibody. Specimens were visualized using an immunofluorescence microscopy (OLYMPUS BX51). Briefly, for α1(V) and α2(V) tridimensional reconstruction by confocal microscopy, the slides were incubated at once with mouse polyclonal antihuman-α1(V) chain (1:300) and rabbit polyclonal antihuman-α2(V) chain (1:80) (Sigma Chemical Co., St. Louis, MO, USA) overnight at 4 °C. Bound antibodies were visualized with Alexa 488-conjugated anti-mouse IgG and Alexa 546-conjugated anti-rabbit IgG (Invitrogen, Eugene, OR, USA). The nuclei were counterstained with 0.4 mM/mL 4’,6-diamidino-2-phenylindole, dihydrochloride (DAPI; Molecular ProbesTM, Invitrogen, Eugene, OR, USA) for 15 min at room temperature. Specimens were observed using a laser scanning microscope (Zeiss LSM 510 META/UV, Oberkochen, Germany). 

### 4.4. Histomorphometry

The intensity of fluorescence corresponding to immunostained α1(V) and α2(V) chains was quantified using image analysis. Briefly, the image analysis system consisted of an Olympus camera (Olympus Co., St. Laurent, QC, Canada) coupled to an Olympus microscope (Olympus BX51), from which the images were sent to an LG monitor by means of a digitizing system (Oculus TCX, Coreco, Inc., St. Laurent, QC, Canada) and downloaded to a computer (Pentium 1330 Mhz). The images were then processed with software Image Pro-Plus 6.0. The area fraction of α1(V) and α2(V) chains was measured in the papillary and reticular dermis and capillary and vessels covering the microscopic field observed at a ×200 magnification. COLV α chain area fraction was expressed as the ratio between the number of measured densities divided by the total area studied in percentage. Ten microscopic fields for each slide were quantified, and the results were expressed as the mean of these fields.

### 4.5. Transmission Electron Microscopy

For transmission electron microscopy, all SSc and control skin samples were fixed in a solution of 2% glutaraldehyde in sodium phosphate and potassium 0.15 M buffer, pH 7. Material was fixed en bloc in aqueous uranyl acetate for 24 h. After this procedure, the samples were dehydrated in acetone and were embedded in Araldite resin. Ultrathin sections were obtained with a diamond knife on LKB Ultratome microtome (Leica, Deerfield, IL, USA). Subsequently, they were placed on copper grid (200-mesh; Lab Research Industries, Burlington, VT, USA) and counterstained with uranyl acetate and lead citrate. Micrographs were taken at 80 kV with transmission electron microscope JEOL 100 cx 100 KW (Philips, Munich, Germany). 

### 4.6. Immunogold Electron Microscopy

SSc and control skin samples were fixed in 2% glutaraldehyde for 2 h at room temperature. Subsequently, the specimens were dehydrated in increasing concentrations of ethanol and embedded in the LR White acrylic resin (London Resin Company, London, UK) for 1 h. After this period, the samples were immersed in gelatin capsules for subsequent polymerization and attainment of the semithin (2 μm) cuts. After staining with toluidine blue to verify the preservation of the histological pattern, the specimens were sectioned in 50 nm thick ultrathin sections and were placed on nickel screen. For ultrastructural immunolocalization by simple indirect labeling, specimens sections were hydrated in water and to block nonspecific sites, incubated with Tris-buffered saline (TBS) containing 0.02 M glycine, pH 7.4 and normal goat serum. Then, the sections were incubated for 12 h at 4 °C with polyclonal mouse antihuman-α1(V) chain (1:30) and polyclonal rabbit antihuman-α2(V) chain (1:30) (Sigma Chemical Co., St. Louis, MO, USA) diluted in TBS containing 0.1% BSA and 0.05% Tween20, pH 7.4. After this time, wash was performed with TBS/BSA for 5 min. Subsequently, the 5 nm colloidal gold-conjugated anti-mouse IgG and 5 nm colloidal gold-conjugated anti-rabbit IgG (Sigma Chemical Co., St. Louis, MO, USA) were diluted 1:50 in 0.5 M NaCl/0.1% BSA with 0.05% Tween20, pH 8.0 and incubated for two hours at room temperature. After washing cycles with TBS/0.1% BSA with 0.05% Tween20, pH 8.0, fixation of the sections was performed in 2.5% glutaraldehyde diluted in 0.1 M sodium cacodylate buffer, pH 7.4. Finally, the sections were washed in distilled water, counterstained with uranyl acetate and lead nitrate and examined using a transmission electron microscope JEOL 100 CX 100 KW (Philips, Munich, Germany).

### 4.7. Cells Culture

The primary fibroblast cell culture was developed using the explant method [41] immediately after the skin biopsy. The SSc and control skin specimens were sectioned in small fragments and transferred to a culture 25 cm^3^ flask (Corning Co., Corning, NY, USA) containing Dulbecco’s modified Eagle’s medium (DMEM) (Gibco, Life Techologies Co., Grand Island, NY, USA) supplemented with 20% fetal bovine serum, 1% L-glutamine and 1% penicillin + streptomycin. The specimens were incubated at 37 °C in humidified atmosphere containing 5% CO_2_ until the confluence of the cells. After 0.2% trypsin in PBS addiction, the cells were transferred to a 75 cm^3^ flask and were maintained at the ideal conditions of temperature and atmosphere as described above. Cells between 2 and 3 passages were used to perform mRNA and protein isolation and *COL5A2* siRNA assays [27]. 

### 4.8. Quantitative Reverse Transcription Polymerase Chain Reaction (RT-qPCR)

For analysis of *COL5A1* and *COL5A2* expression, once the skin fibroblasts from SSc patients and controls reached confluence (100%) in 75 cm^3^ flask, total RNA was isolated according to a standard Trizol^®^ (Invitrogen, Life Techologies Co., Carlsbad, CA, USA) RNA isolation protocol. Samples of total RNA were quantified by measuring the optical density (Nano Vue Plus^®^ Spectrophotometer; GE, Biochrom LTD, Cambridge, UK). Gene expression was determined using real time reverse transcriptase polymerase chain reaction (RT-PCR) analysis with ACTA as housekeeping gene. All reverse transcription reaction mixtures were prepared using Superscript Platinum III One-Step kits (Invitrogen, Life Techologies Co., Carlsbad, CA, USA) and performed on Step One^®^ thermocycler (Applied Biosystems, Life Techologies Co., Carlsbad, CA, USA). The cDNA synthesis was performed at 50 °C for 10 min. After that, RT-PCR reactions were performed with the primers shown in Table 2. 

The sequences of the genes were acquired by (www.ncbi.nem.nih.gov/nucleotide accessed on 30 August 2022) site. For assembling the gene map of the chosen sequence, software located at the electronic address (www.genome.ucsc.Edu/cgi.bin/hgbeat accessed on 30 August 2022) site was used.

The RT-PCR conditions were 95 °C for 30 s, 60 °C for 30 s and 72 °C for 1 min for 35 cycles. The control group samples were used as calibrator, and the relative expression was calculated with 2^−ΔΔCT^ method. Analysis of relative gene expression data was performed using real-time quantitative PCR and the 2[−Delta-Delta C(T)] method [42]. 

### 4.9. Western Blotting

SSc and control dermal fibroblasts were harvested and lysed in NP-40 cell lysis buffer (Invitrogen), containing PMSF 1 mM (Sigma Chemical Co., St. Louis, MO, USA) and protease inhibitor cocktail (Sigma Chemical Co., St. Louis, MO, USA), according to manufacturer protocol. Cell extracts (30 µg) were separated using electrophoresis in sodium dodecyl sulfate polyacrylamide gels 10% under reducing conditions, as previously described [43]. Separated proteins were electroblotted onto nitrocellulose membranes (Amersham, GE Healthcare Life Sciences, Chicago, IL, USA), and the membranes were blocked for 90 min at room temperature in PBS/5% bovine serum albumin/0.1% Tween20. The transferred proteins reacted for 18–20 h at 4 °C with mouse polyclonal antihuman-α1(V) chain (1:1000), rabbit polyclonal antihuman-α2(V) chain (1:200) (Sigma Chemical Co., St. Louis, MO, USA) and β-actin (1:1000) (Sigma Chemical Co., St. Louis, MO, USA). After incubation with horseradish peroxidase–conjugated anti-rabbit or anti-mouse IgG antibodies (Sigma Chemical Co., St. Louis, MO, USA) for 1 h, signals were detected with an enhanced chemiluminescence detection system (ECL Plus; Amersham, GE Healthcare Life Sciences, Chicago, IL, USA). Densitometry quantification was performed using Image J software, and signals were normalized to those of β-actin. 

### 4.10. siRNA Synthesis and Transfection

The fibroblasts were transiently transfected with *COL5A2* small interfering RNA (siRNA; TriFECTaTM kit Integrated DNA Technologies—RTD) using Lipofectamine 2000 reagent (Invitrogen, Life Techologies Co., Carlsbad, CA, USA). siRNA synthesized from an irrelevant sequence (DS scrambled negative) without homology to human genes was used as a negative control and methodology and HPRT siRNA-S1, commonly found in human gene, was used as positive control in accordance with the manufacturer instructions. Briefly, the siRNA mixture (10 mM) and Lipofectamine 2000 were mixed in serum-free DMEM without antibiotics and added to the cells, which were then harvested 72 h after transfection.

### 4.11. Statistical Analysis

Statistical analysis was performed using the statistical package for the Social Sciences version 20.0 (SPSS, Chicago, IL, USA). All data were expressed as mean and standard deviations for continuous variables and percentages for categorical variables. The regular distribution of the data in the groups was tested using the Kolmogorov–Smirnov test. The Mann–Whitney test was used to compare unpaired and nonparametric continuous variables and the student t-test for paired and parametric continuous variables. For significant differences between groups for a given variable, we used Pearson’s correlation coefficient for the distribution of parametric data and Spearman for the distribution of nonparametric data. The level of significance was *p* < 0.05.

## Figures and Tables

**Figure 1 ijms-23-12654-f001:**
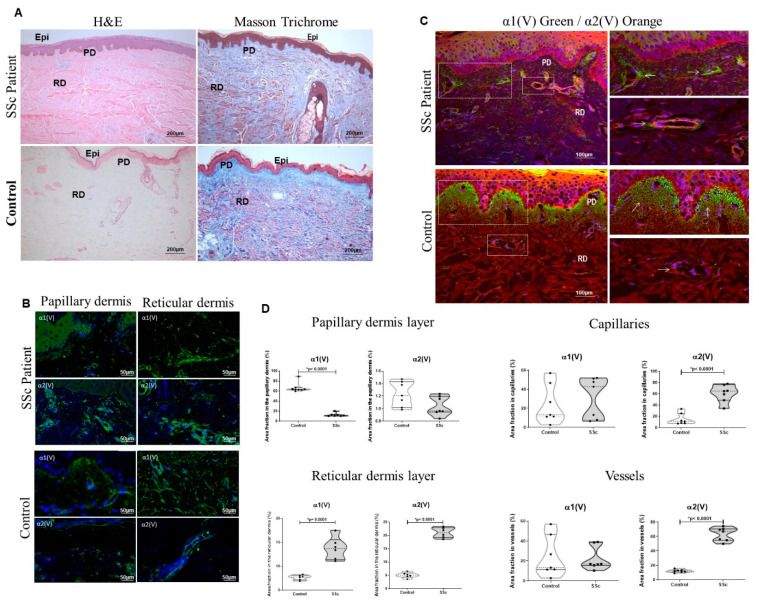
Expression of α1(V) and α2(V) chains in early-SSc patient skin. (**A**) Histologic features using H&E and Masson trichrome staining. Immunofluorescence showing the α1(V) and α2(V) chain expression in papillary and reticular dermis (green; (**B**)) and the colocalization of the α1(V) and α2(V) chains in the SSc and control, respectively, visualized in green and red (**C**). PD: papillary dermis; RD: reticular dermis. (**D**) Graphic representation of the area fraction of the α1(V) and α2(V) chains present in papillary and reticular dermis, capillaries and vessels. Note the significant decrease in α1(V) in SSc-papillary dermis and increased α2(V) expression in capillaries and vessels in the SSc skin in relation to control. GraphPad Prism 8.0 software: *t*-test, *p* < 0.0001 or *p* = 0.001 (*).

**Figure 2 ijms-23-12654-f002:**
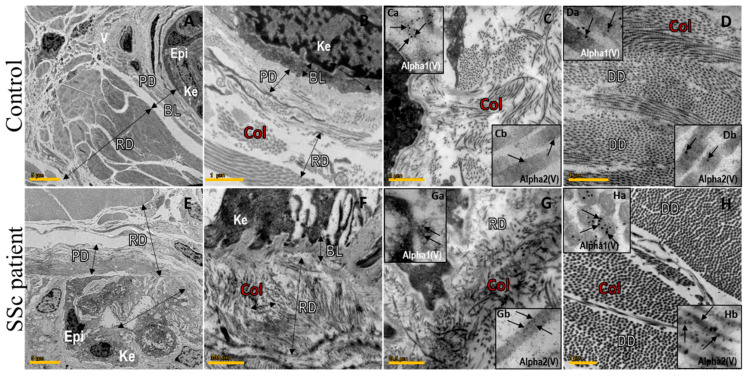
Representative ultrastructural photomicrographs of skin from early-SSc patients and control showing the fibrillar structure and immunogold stained α1(V) and α2(V) chains. (**A**) Control skin with preserved Epi, PD and RD architecture and homogeneous collagen distribution. (**B**–**D**) Normal delimitation of BL, PD and RD and normal collagen (Col) fibril architecture. (**E**,**F**) SSc skin exhibits rectification of epidermis, disarray and prominence of Col fine fibrils determining significant thickness and fusion of BL with RD. (**G**,**H**) Disarray and prominence of Col thick fibrils determining significant thickness of RD and DD. Inserts in C-D and G-H show α1(V) and α2(V) chain immunogold labeled in control and SSc, respectively. The SSc epidermal–dermal junction shows decreased α1(V) (Ga) compared to control (Ca). A strong α1(V) and α2(V) expression is shown joining the thickened region of DD (Ha, Hb) from SSc skin in contrast with control (Da, Db). Epi: epidermis; PD: papillary dermis; RD: reticular dermis; Ke: keratinocytes; BL: basal lamina; DD: depth dermis; and Col: collagen. Scale Bars: (**A**,**E**) 5 µm; (**B**–**D**) 1 µm; (**F**,**H**) 100 nm; (**G**) 0.5 µm.

**Figure 3 ijms-23-12654-f003:**
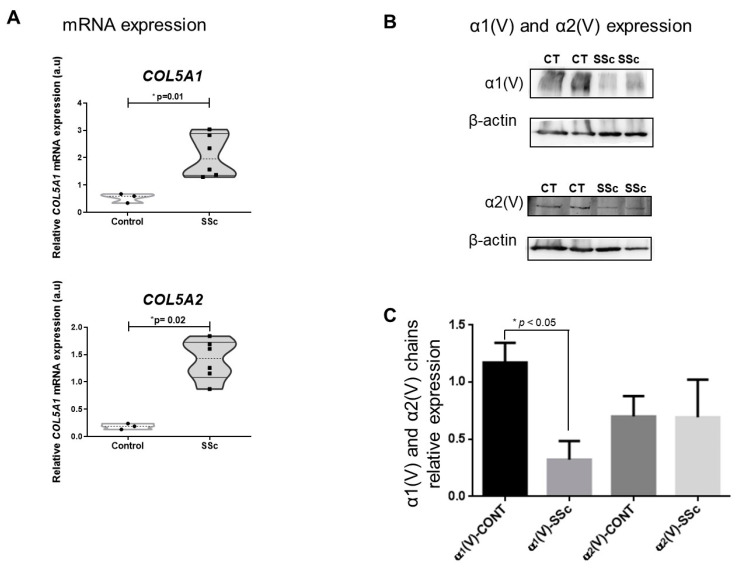
Gene and protein expression of α1(V) and α2(V) chains in the skin fibroblasts of the early-SSc patients and control. (**A**) Relative expression of *COL5A1* and *COL5A2* mRNA in SSc and control. (**B**) Immunoblotting shows α1(V) and α2(V) chain expression in skin fibroblasts from SSc and control (CT). (**C**) Relative expression of α1(V) and α2(V) chains from SSc and control cutaneous fibroblasts was evaluated using the β-actin endogenous control, loaded in the same gels of the analyzed samples. GraphPad Prism 8.0 software: t-test: *p* < 0.05 (*).

**Figure 4 ijms-23-12654-f004:**
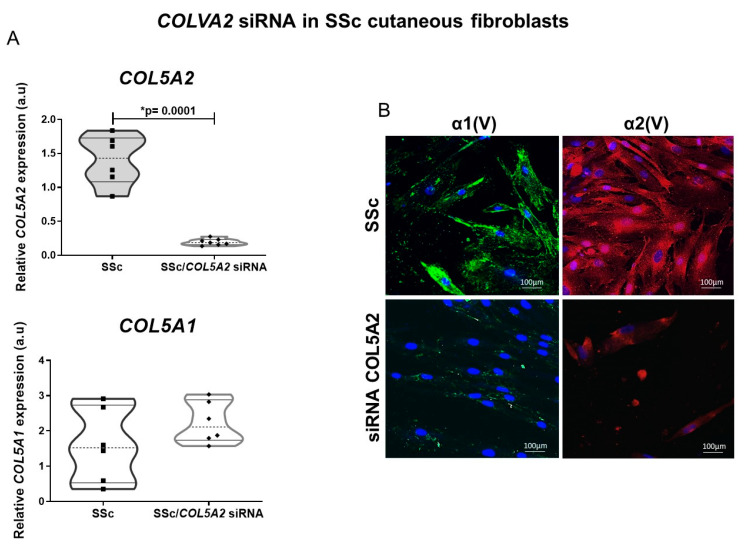
Expression profile of COLVA2 siRNA in SSc cutaneous fibroblast efficiency. (**A**) mRNA expression of *COL5A2* was significantly inhibited after transfection with specific siRNA. (**B**) Immunofluorescence confirmed down regulation of α2(V) and α1(V) protein levels in cells transfected with siRNA. * Statistical significance.

**Table 1 ijms-23-12654-t001:** Clinical and laboratory characteristics of patients.

Clinical and Laboratory Features	
N (%)	7 (100%)
Age, standard deviation (range)	42 (12)
Sex, N (%)	Female, 4 (60%) ^1^ Male, 3 (40%)
Clinical subtype, N (%)	Limited, 3, (40%) Diffuse, 4 (60%)
Disease duration, median (range)	12 (14–24 meses)
MRSS^1^, median (range)	25 (11–40)
Valentini activity index ^2^, median (range)	0.5
FAN+, N (%)	7 (100%)
Anti-Scl70+, N (%)	2 (28.7%)
Anticentromere+, N (%)	4 (51%)

^1^ Modified Rodnan Skin Score. ^2^ According to the Valentini activity index^40^. Regarding the histological diagnosis of the patients, all had mild fibrosis (100%).

**Table 2 ijms-23-12654-t002:** Sequence of oligonucleotides.

Gene	Sense 3′—5′	Antisense 5′—3′	(pb)
ACTB	AGAAAATCTGGCACCACACC	AGAGGCGTACAGGGATAGCA	175
COL5A1	GTGGCACAGAATTGCTCTCA	CTGGATGTCACCCTCAAACA	175
COL5A2	GGAAATGTGGGCAAGACTGT	TTGATGGTGGTGCTCATTGT	169

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
