# Peer review of "Collagen V α1 Chain Decrease in Papillary Dermis from Early Systemic Sclerosis: A New Proposal in Cutaneous Fibrosis Molecular Structure"

_ijms, 2022, doi:10.3390/ijms232012654_

Round 1
Reviewer 1 Report
Reference 2 needs supplementing with Tsou PS et al Advances in epigenetics in systemic sclerosis: molecular mechanisms and therapeutic potential Nature Reviews Rheumatology 2021 17: 596
Is it possible to quantify the changes demonstrated in figure 2 in the electron microscope images? Is this quantifiable and if so please show a graph of this?
The authors show a downregulation of alpha 1 (V) chain in systemic sclerosis patients by western blotting. Whilst this is convincing the authors must use healthy dermal fibroblasts control cells and treat (and compare to non treated cells) with the molecule TGF-b1 (2 ng/ml) and then measure by WB the alpha 1 (V) chain to determine if TGF beta 1 is the mediator important in this selective repression.
Figure 4 not sure what is trying to be made point of here?? Does the siRNA reduce formation of myofibroblasts?
Author Response
First of all, we would like to thank to both Reviewers for the critical comments that allowed us to revise the work to make it more fluent and clearer to justify the rationale of the study and minimize major weaknesses in regard to data presentation and interpretation. We revised the manuscript based on his comments and highlighted the answers to queries in yellow, and we are re-submitting the paper along with a point-by-point response to the reviewer' comments. Two other prerogatives need to be highlighted: 1) the respect and consideration for the patients included in the study and the benefits they will bring to other patients, and 2) the urgent establishment of therapeutic strategies and the clinical outcomes of patients with devastating disease like Progressive Systemic Sclerosis. The intellectual exercise carried out between authors and reviewers undoubtedly increased the scientific value of the work. Thank you very much.
Response to Reviewer 1 Comments
Point 1: Reference 2 needs supplementing with Tsou PS et al Advances in epigenetics in systemic sclerosis: molecular mechanisms and therapeutic potential Nature Reviews Rheumatology 2021 17: 596.
Response 1: We greatly appreciate your suggestion. The reference has been included in the introduction to the manuscript. Thank you very much for your comment that helped to improve our manuscript.
Point 2: Is it possible to quantify the changes demonstrated in figure 2 in the electron microscope images? Is this quantifiable and if so please show a graph of this?
Response 2: We appreciated this suggestion very much. Unfortunately, the quantification of immunogold particles requires a highly sensitive immunoelectron microscopy technique called sodium dodecyl sulfate-digested freeze-fracture replica immunogold labeling (Fujimoto K. (1995). Freeze-fracture replica electron microscopy combined with SDS digestion for cytochemical labeling of integral membrane proteins. Application to the immunogold labeling of intercellular junctional complexes. J. Cell Sci. 108 3443–3449. 10.1242/jcs.108.11.3443), multiple collagen proteins can be tagged with different sizes of immunogold particles at once and visualized two-dimensionally. For quantification, gold particles the images must be annotated, and then different mathematical and statistical methods must be applied to characterize the distribution states of proteins interest. To perform such analyses, it is necessary a special program, which integrates several classical and novel analysis methods for immunogold labeled replicas into one self-contained package. The program not only performs the selected analysis but also automatically compares the results of the real distribution to a random distribution of the same number of particles on the collagen fibers of interest. Thanks a lot for the suggestion.
Point 3: The authors show a downregulation of alpha 1 (V) chain in systemic sclerosis patients by western blotting. Whilst this is convincing the authors must use healthy dermal fibroblasts control cells and treat (and compare to non treated cells) with the molecule TGF-b1 (2 ng/ml) and then measure by WB the alpha 1 (V) chain to determine if TGF beta 1 is the mediator important in this selective repression.
Response 3: We totally agree with the Reviewer for the suggestion. However, the Reviewer must agree that to explore such mechanistic, we will need new samples from patients with SSc and healthy volunteers. Therefore, we are committed to carrying out further studies following the Reviewer's suggestion. Thank you for the careful analysis of our manuscript.
Point 4: Figure 4 not sure what is trying to be made point of here?? Does the siRNA reduce formation of myofibroblasts?
Response 4: We agree with the referee comment. The objective of partial silencing of the α2(V) chain was to decrease the expression of this collagen chain, however we did not observe cell decrease in our study. Actually, what was represented in the image of the figure was the decrease in protein expression. Thank you very much for your comment.

Reviewer 2 Report
De Morais et al. provided evidence that the decrease concentration of a1(V) chain in the papillary dermis of SSc skin sample influences the biomechanical properties contributing thickening and cutaneous fibrosis in scleroderma.
The experimental plan is well designed and the data clearly shown.
Major concerns:
- Characteristics of patients: the authors affirm that are early-SSc but the table shows limited and diffuse form. So the patients are not so "early". Please specify.
- In Fig 3 and 4 the authors provide data showing a1(V) and a2(V) expression in SSc and normal cells. Did the authors obtained histologic sections and fibroblasts from the same patient or controls? These data are from the same patients or from different patients? The patients have limited or diffuse SSc?
- Are there any correlation between col V and disease activity or other clinical characteristics?
- Since the author analyzed a1(V) and a2(V) expression in papillary and reticular dermis by histological observation and in fibroblasts obtained from whole skin sample, it would be important to perform a single cell analys in the 2 different area.
- Which kind of cells are the one positive for a1(V) and a2(V) expression ?
Based on the nature of the findings it is difficult think that as the authors affirm, that "col V role is important to determined new target for future treatment" . please give further interpretation and the significance of the results in the discussion section.
Author Response
First of all, we would like to thank to both Reviewers for the critical comments that allowed us to revise the work to make it more fluent and clearer to justify the rationale of the study and minimize major weaknesses in regard to data presentation and interpretation. We revised the manuscript based on his comments and highlighted the answers to queries in yellow, and we are re-submitting the paper along with a point-by-point response to the reviewer' comments. Two other prerogatives need to be highlighted: 1) the respect and consideration for the patients included in the study and the benefits they will bring to other patients, and 2) the urgent establishment of therapeutic strategies and the clinical outcomes of patients with devastating disease like Progressive Systemic Sclerosis. The intellectual exercise carried out between authors and reviewers undoubtedly increased the scientific value of the work. Thank you very much.
Response to Reviewer 2 Comments
Comments and Suggestions for Authors
De Morais et al. provided evidence that the decrease concentration of a1(V) chain in the papillary dermis of SSc skin sample influences the biomechanical properties contributing thickening and cutaneous fibrosis in scleroderma.
The experimental plan is well designed, and the data clearly shown.
Major concerns:
Point 1: Characteristics of patients: the authors affirm that are early-SSc but the table shows limited and diffuse form. So, the patients are not so "early". Please specify.
Response 1: We totally agree with the Reviewer. In this manuscript version, the characterization of the patients was not very well clarified. In the new version of the manuscript we clarify that, we chose to select only patients who were diagnosed with SSc within the first two years since the first non-Raynaud´s symptom and who were treatment-naïve without immunesuppressors and/or antifibrotics, as explained in the discussion. Our goal was to understand the role of the Col V in the first two years of the skin fibrosis in SSc, who is considered a key moment in the treatment of SSc cutaneous fibrosis. Thanks so much to the careful evaluation of our paper.
Point 2: In Fig 3 and 4 the authors provide data showing a1(V) and a2(V) expression in SSc and normal cells. Did the authors obtain histologic sections and fibroblasts from the same patient or controls? These data are from the same patients or from different patients? The patients have limited or diffuse SSc?
Response 2: We appreciate this question very much. The histologic sections and the fibroblasts were collected from the same patients. Control skin and fibroblasts were obtained of the same subjects and the biopsies were realized from the same region of SSc patients in elective surgery procedures. The presented data were from the same patients. Concerning to disease clinical subsets, we investigated all skin biopsies and fibroblasts from the Limited (n=3) and Diffuse (n=4) forms. We would like to clarify that fibroblasts isolated from these two forms of the disease have already been shown to share many of the main features at the epigenomic level (Altorok N et al. Genome-wide DNA methylation analysis in dermal fibroblasts from patients with diffuse and limited systemic sclerosis reveals common and subset-specific DNA methylation aberrancies. Ann Rheum Dis. 2015; 74(8): 1612–1620). In fact, we found no difference between the histological characteristics of the skin and in the expression of α1(V) and α1(V)2 chains between the subtypes of SSc patients. Thank you for the very pertinent question.
Point 3: Are there any correlation between col V and disease activity or other clinical characteristics?
Response 3: This is really a great question. In our previous work, we demonstrated abnormal synthesis of Col V in SSc skin biopsies and its correlation with disease stage, disease activity using the Valentini Disease Activity Index and skin thickening using the Modified Rodnan Skin Score (MRSS) (Martin P et al. Abnormal collagen V deposition in dermis correlates with skin thickening and disease activity in systemic sclerosis. Autoimmun Rev. 2012, 11,827–35). As mentioned in our manuscript, Col V is a fibrillar protein made up of three α chains, which a homotrimer isoform is composed of three α1(V) and a heterotrimer isoforms, is composed of the two α1(V) chains and one α2(V) chain. In the present study, our goal was to evaluate the morphological, ultrastructural and molecular features of α1(V) and α2(V) chains of the collagen V in skin biopsies and fibroblasts in a cohort of SSc patients with initial manifestations of the disease and treatment-naïve and, whereas our series of patients, although rare, is small and unfortunately, we do not correlate the expression of alpha chains in different regions of the skin with disease activity or skin thickening. This is a limitation of the present study and can be the subject of study for the next manuscript.
Point 4: Since the author analyzed a1(V) and a2(V) expression in papillary and reticular dermis by histological observation and in fibroblasts obtained from whole skin sample, it would be important to perform a single cell analysis in the 2 different area.
Response 4. We totally agree with the Reviewer for the suggestion. We performed the analysis in the two different areas of the skin, but we don’t evaluated fibroblasts isolated from papillary and reticular regions of the dermis. Again, it is a limitation of the study and to analyze a single cell isolated from these regions, we will need new samples from patients with SSc in the first two years of disease treatment-naïve without immunosuppressors and/or antifibrotics and healthy volunteers. Therefore, we are committed to carrying out further studies following the Reviewer's suggestion. Thank you for the careful analysis of our manuscript and valuable suggestions and valuable suggestions that will improve our line of research.
Point 5: Which kind of cells are the one positive for a1(V) and a2(V) expression?
Response 5. Again, this is a very interesting question. Unfortunately, we currently do not have specific markers for the molecular phenotype of the papillary (NTN1, PDPN, ACKR4), reticular (TGM2, CNN1, CDH2, MGP) and dermal-epidermal junction fibroblasts, as well as undifferentiated cells (CD271), for immunostaining and colocalization assays. Indeed, this is another limitation of the present study and we are committed to carrying out further studies following the Reviewer's suggestion. Thank you for the careful analysis of our manuscript.
Point 6: Based on the nature of the findings it is difficult think that as the authors affirm, that "col V role is important to determined new target for future treatment”. Please give further interpretation and the significance of the results in the discussion section.
Response 6. Again, the referee is completely right. In the present version of the manuscript, this idea was not clear and our results are not sufficient to support this suggestion. Therefore, in the new version of the manuscript we have rewritten this paragraph, as the referee can see below. Thank you very much for your comment that helper to improve our manuscript.
“….considering that atypical expression of the α1(V) chain may interfere with skin thickening, we can infer that α1(V) chain expression regulation could be a possible therapeutic target at the onset of cutaneous fibrosis in SSc”

Round 2
Reviewer 1 Report
References 35 and 36 need supplementing with Henderson J et al
The Role of Epigenetic Modifications in Systemic Sclerosis: A Druggable Target Trends in molecular medicine 2019 25.
Author Response
Response to Reviewer 1 Comments
Comments and Suggestions for Authors
References 35 and 36 need supplementing with Henderson J et al. The Role of Epigenetic Modifications in Systemic Sclerosis: A Druggable Target Trends in molecular medicine 2019 25.
Response: Again, we appreciate your suggestion that helped to improve our manuscript. As you suggested the reference was included in the manuscript discussion. Thank you very much for your comment.

Reviewer 2 Report
The authors have improved their manuscript replying to the reviewer questions
Author Response
Response to Reviewer 2 Comments
The authors have improved their manuscript replying to the reviewer questions
Response: We appreciate the reviewer's suggestions that helped to improve our manuscript.
